# Features influencing the health and economic impact of preventing COVID-19 in immunocompromised individuals

Siyu Chen[1], Sudhir Venkatesan[2], Sofie Arnetorp[3], Klas Bergenheim[3], Sabada Dube[4], Wilhelmine Meeraus[4], Cátia Ferreira[5], Sylvia Taylor[6], Lisa J. White[7,8]*

1 Department of Public and Ecosystem Health, College of Veterinary Medicine, Cornell University, Ithaca, New York, United States of America, 2 Medical and Payer Evidence, BioPharmaceuticals Medical, AstraZeneca, Cambridge, United Kingdom, 3 Health Economics & Payer Evidence, BioPharmaceuticals R.&D., AstraZeneca, Gothenburg, Sweden, 4 Medical Evidence, Vaccines and Immune Therapies Unit, AstraZeneca, Cambridge, United Kingdom, 5 Vaccines and Immune Therapies, BioPharmaceuticals Medical, AstraZeneca, Wilmington, Delaware, United States of America, 6 Self-employed, Cambridge, United Kingdom, 7 Model Health Ltd, Oxford, United Kingdom, 8 Department of Biology, University of Oxford, Oxford, United Kingdom

* lisa.white@modelhealth.online, lisa.white@biology.ox.ac.uk

## Abstract

Many immunocompromised individuals mount inadequate immune responses following COVID-19 vaccination, thus relying on other social distancing behaviours, particularly shielding, for protection, impacting their quality of life. However, little is known about historical/current levels and effectiveness of shielding or factors influencing individuals' decision to continue shielding. Long-acting antibody pre-exposure prophylaxis (LAAB-PrEP) provides direct protection against COVID-19 in immunocompromised individuals who have been and may continue to shield. However, the proportion and incidence of circulating variants for which LAAB-PrEP would be effective is unpredictable. Given this uncertain behavioural and immuno-epidemiological context, we developed a modelling framework to explore features that most impact health outcomes and cost effectiveness of long-term administration of LAAB-PrEP against COVID-19 infection in immunocompromised individuals in the English context. The model predicted that the incremental cost-effectiveness ratio (ICER) of LAAB-PrEP against COVID-19 in immunocompromised individuals will be largely driven by features of utility of shielding, current/future shielding behaviour, cost of shielding, risk of COVID-19 hospitalisation among immunocompromised individuals and the time horizon used for the cost-effectiveness analysis. The model estimated that for realistic ranges of influential factors, it is possible for LAAB-PrEP to be cost effective under the conditions that most immunocompromised individuals would shield indefinitely if it were not available but would switch to LAAB-PrEP if it were. Thus, if individuals stop shielding when taking LAAB-PrEP, then LAAB-PrEP is cost effective.

**Data availability statement:** All data, code and materials used in the analyses can be accessed at: https://github.com/SiyuChenOxf/Modelling-for-CEA-shielding-and-LAAB. All parameter estimates and figures presented can be reproduced using the code provided.

**Funding:** This work was conducted at the University of Oxford with partial funding from the AstraZeneca-sponsored RAVEN study (EUPAS43571; NCT05047822 to LJW and SC). The funders had no role in study design, data collection and analysis, decision to publish, or preparation of the manuscript.

**Competing interests:** I have read the journal's policy and the authors of this manuscript have the following competing interests: SV, WM, KB, SA, SD and CF are employees of AstraZeneca and hold AstraZeneca stocks/shares. ST is a former employee of AstraZeneca. LJW is the director of Model Health Ltd, a company that provides paid consultancy services to AstraZeneca.

## Author summary

Vaccinations, including vaccinations against COVID-19, may not be very effective in people with weakened immune systems. This is because these immuno-compromised individuals may not mount an adequate immune response to a vaccine. Individuals who are immunocompromised are at greater risk of severe COVID-19 than those who are not immunocompromised. It is therefore recommended that these individuals shield themselves, only going out when essential. However, this shielding behaviour reduces these individuals' quality of life. A possible solution is to inject immunocompromised individuals with long-acting antibodies against COVID-19. Unfortunately, the level of protection gained from both shielding and long-acting antibodies is uncertain and comes at a high cost, financially and/or to immunocompromised individuals' quality of life. Here, we used a theoretical, mathematical modelling approach to explore which factors might affect the health and economic impacts of these long-acting antibodies in various scenarios. By taking shielding behaviour into consideration, our model suggested that the expected health and economic impacts of long-acting antibodies for COVID-19 in individuals who are immunocompromised will be driven by the features of their shielding behaviour, how likely they are to die due to the underlying cause of their weakened immune system, how likely they are to be hospitalised due to their COVID-19 infection, and the effectiveness of long-acting antibodies in preventing clinically significant disease. Furthermore, our model suggested that if LAAB-PrEP enables immunocompromised individuals to stop shielding, it would be cost effective.

## Introduction

The COVID-19 pandemic instigated an unprecedented global response, including huge efforts to develop pharmaceuticals for prophylaxis against SARS-CoV-2. The COVID-19 vaccines, delivered as regular booster doses, are highly effective at reducing disease severity and mortality [1,2]. However, some individuals, such as those who are immunocompromised, may mount an inadequate immune response to COVID-19 vaccines [3], leaving them at risk of COVID-19. Immunocompromised individuals comprise 2–7% of the global population [4] and, regardless of their vaccination status, during the pandemic they remained at substantially greater risk of severe or fatal COVID-19 than their non-immunocompromised counterparts [5]. This elevated relative risk continued as the pandemic began to transition to the endemic phase [6]. There is an ongoing unmet need for prophylactic therapies against SARS-CoV-2 for immunocompromised and other individuals at high risk of severe COVID-19 infection [7].

Various non-pharmaceutical interventions were introduced during the pandemic in an attempt to limit the spread of COVID-19, including social distancing, face masks and lockdowns. Some non-pharmaceutical interventions, such as

shielding, were specifically recommended for immunocompromised and other clinically vulnerable individuals to protect themselves from infection. Shielding refers to individuals' self-quarantining behaviours, only going out when essential (for more details, see the Shielding section of [8]). Through the various stages of the pandemic, immuno-compromised individuals were advised to continue to shield at varying degrees of intensity to reduce their risk of SARS-CoV-2 infection and severe outcomes [9,10]. However, this practice can negatively impact individuals' mental health and health-related quality of life [9,10]. As with any intervention, shielding therefore has costs and disutility, both to individuals and the state (see Tables A and B in S1 Text). Costs to states during the COVID-19 pandemic included financial support provided to the population by governments, productivity losses due to shielding preventing work that could not be performed from home, and hospitalisation costs of breakthrough infections due to shielding not being 100% effective. Costs to individuals included the potential loss of income, while disutility included social isolation, an inability to lead a normal life, a feeling among some shielding individuals of being 'second-class citizens' and 'left behind' as restrictions ended for the rest of the population [11] and increased anxiety and symptoms of depression [9,12].

Various pharmaceutical therapies for treatment or prophylaxis are available or under development to reduce the incidence and severity of COVID-19 infection among individuals who continue to shield. Antiviral drugs, such as remdesivir, can be used to treat COVID-19 [13], while monoclonal antibodies (mABs) that neutralise SARS-CoV-2 can be used for prophylaxis (e.g., bamlanivimab monotherapy [14]) or treatment (e.g., bamlanivimab and etesevimab combination therapy [15]). However, the half-lives of these mABs are typically short (18–32 days), necessitating monthly administration, sometimes intravenously [16]. Efforts are therefore increasingly focused on long-acting antibodies (LAABs), which offer protection of duration around six months, such as AstraZeneca's *Evusheld*, a combination of LAABs comprising tixagevimab plus cilgavimab [3]. When used for pre-exposure prophylaxis (PrEP) in 2021, one dose of *Evusheld* reduced recipients' risk of COVID-19 symptomatic infection and hospitalisation by 82.8% (95% CI, 65.8 to 91.4) for at least six months [16], reflecting high effectiveness against the variants circulating in the study population at the time of the study. LAABs continue to be updated to maintain their effectiveness as the variant landscape changes over time. For example, an investigational LAAB (sipavibart, formerly AZD3152) [17] continues to be updated for effectiveness and is undergoing clinical trials [18]. Evaluating mAb efficacies, and to what extent the effectiveness of a given mAb contributes to its cost effectiveness when used prophylactically, remains challenging. An added complication is that whereas previously there were sequentially dominant variants [19], there are now cocirculating variants with temporally differing degrees of dominance [20].

As potentially effective new prophylactic LAAB therapies emerge, public health planners need ways to evaluate their potential impact by comparing their advantages with current approaches such as shielding, to select the most cost-effective approach. Potential health and economic impacts to consider include increased life expectancy among those receiving long-acting antibody pre-exposure prophylaxis (LAAB-PrEP), reduced demand for hospital resources and broader economic impacts.

Multiple features may contribute to the cost effectiveness of LAAB-PrEP, including those associated with the existing and possible future immuno-epidemiological landscape, the current and future shielding behaviour of immunocompromised individuals, the selection of primary conditions that individuals considered eligible for LAAB-PrEP at any given time have, and the time horizon that decision-makers are considering. Many of these features vary over time, have not been accurately measured or both. For this study, therefore, we were not seeking to determine whether the hypothetical pharmaceutical intervention (LAAB-PrEP), in combination with shielding, is cost effective as there are so many uncertainties around both the intervention itself and human behaviour, although our model could be used for this purpose if more accurate input values were available. We instead used the model to explore which features of the risk group, LAAB-PrEP effectiveness, shielding effectiveness, and cost most influence the health and economic impacts of LAAB-PrEP under various feasible epidemiological and behavioural scenarios. We describe the model's inputs, outputs, computational strategies, interpretation, implications and limitations.

The model was developed to assess the potential impact of LAAB-PrEP when administered to individuals who are immunocompromised under various feasible epidemiological and behavioural scenarios in the high-income context of England. To evaluate the potential impact of any newly licensed pharmaceutical intervention, the intervention must be assessed against the existing standard of care as a comparator. However, the standard of care to protect immunocompromised individuals against COVID-19 is shielding, a non-pharmaceutical intervention that has costs to the individual (and their families) [7], but not necessarily always to the state/government, and which are rarely quantified. Unifying pharmaceutical and non-pharmaceutical interventions into a single cost-effectiveness evaluation framework is challenging. The analysis has to account for the fact that LAAB-PrEP is unlikely to be associated with complete replacement of shielding, but rather a continuum of differing proportions of the two. In our analysis, our intervention must be defined as a combination of a pharmaceutical intervention (LAAB-PrEP) and a non-pharmaceutical intervention (shielding), with the latter expected to continue, albeit at a possibly reduced level [7]. The pharmaceutical element of the intervention (LAAB-PrEP) is subject to rigorous safety and effectiveness standards in addition to cost-effectiveness analysis with an identified payer. Our comparator, however (shielding, which is also, at a different level, a part of the intervention combination) is a non-pharmaceutical intervention that has been implemented without requiring the same standards of safety or effectiveness assessment. Furthermore, shielding was implemented with limited consideration of its cost effectiveness and with the cost generally being covered by individuals themselves. Other behavioural factors could influence the extent to which individuals stop shielding given the availability or not of LAAB-PrEP; for example, perceptions of the severity and risk of COVID-19 and confidence in LAAB-PrEP to provide sufficient protection would influence such decisions.

An economic model is one way to address this challenge, enabling comparison of our comparator (shielding at current or reduced levels) with our hypothetical intervention (LAAB-PrEP combined with a current or reduced level of shielding). Here, we describe a novel modelling framework to explore which features most impact the cost effectiveness of the long-term (repeated doses over a 5- to 10-year time-horizon) administration of hypothetical LAAB-PrEP against COVID-19 infection in individuals who are immunocompromised. For such a model to be useful and aid decision-making, it must be able to evaluate the relative advantages and disadvantages of the comparator and the intervention in terms of incremental cost-effectiveness [21].

Another factor we must consider is reinfection. This is an important aspect of COVID-19 epidemiology, particularly among individuals who are immunocompromised, because of their weakened immune system and/or suboptimal response to COVID-19 vaccines [22]. It is reasonable to assume therefore that individuals will undergo repeated challenges, increasing their risk of severe disease or death, especially for those individuals who do not mount a natural immune response to such challenges. In addition, it is likely that protection from any hypothetical LAAB-PrEP will wane over time, so repeated dosing at fixed intervals for all target groups would be required to maintain long-term protection. We therefore also consider repeat dosing strategies for high-risk individuals in this analysis.

We explored various scenarios by conducting a large-scale sensitivity analysis in which various comparators and a range of interventions were sampled and compared to examine their cost effectiveness. The ranges of input values for the model are consistent with the epidemiological and behavioural setting in England.

## Results

### Comparing shielding with the hypothetical pharmaceutical intervention (LAAB-PrEP) combined with potentially reduced shielding

We begin by comparing a scenario in which a group of immunocompromised individuals, the target population, are following the current standard of care, i.e., shielding (our comparator) with a scenario in which an alternative group of individuals in the target population to whom the hypothetical LAAB-PrEP in combination with shielding (our intervention) is available. We assume that the availability of LAAB-PrEP will result in individuals ceasing to shield to varying and unpredictable degrees [7], with different levels of effect in terms of quality-adjusted life-years (QALYs) gained. QALYs are

PLOS Computational Biology

a measure of disease burden that incorporate the quantity and quality of life and are routinely used in health economics [23]. Fig 1 shows a dot plot of incremental cost ($\Delta C$) versus incremental QALYs ($\Delta Q$) from the healthcare payer perspective where the cost of shielding is not considered (see also Fig D in S1 Text where the cost of shielding is considered). All of the model's parameters are randomly sampled from all feasible ranges (the parameters and ranges are shown in Tables A-D in S1 Text). We find that for nearly all of the scenarios, there is an increase in QALYs with the combination of hypothetical LAAB-PrEP with shielding versus the standard of care. The predicted probability of cost effectiveness of hypothetical LAAB-PrEP combined with shielding is 0.1 from the healthcare payer perspective given that the willingness-to-pay threshold is 20,000 GBP; this increases to approximately 0.6 if the willingness-to-pay threshold is 100,000 GBP [24].

**Key features influencing the health and economic impact of LAAB-PrEP**

We examined the influence of each feature on incremental cost-effectiveness ratios (ICERs), incremental QALYs ($\Delta Q$) and incremental costs ($\Delta C$), with time horizons of 2, 5 and 10 years, using a one-way sensitivity analysis, visualised using tornado plots; this was from the healthcare payer perspective where shielding cost was not considered (Fig 2; see also Fig G in S1 Text for a correlation analysis between QALYs and features and Fig F in S1 Text where shielding cost is considered). Features influence the health and economic impact of LAAB-PrEP measured using these three metrics in different directions and in a different order of importance. For example, increasing shielding replacement ($\rho$) would lead to an increasing incremental cost but a decreasing incremental QALY, eventually resulting in a decreasing ICER. The unit cost of LAAB-PrEP is ranked as the most important feature determining the incremental costs, while the utility of shielding tends to be

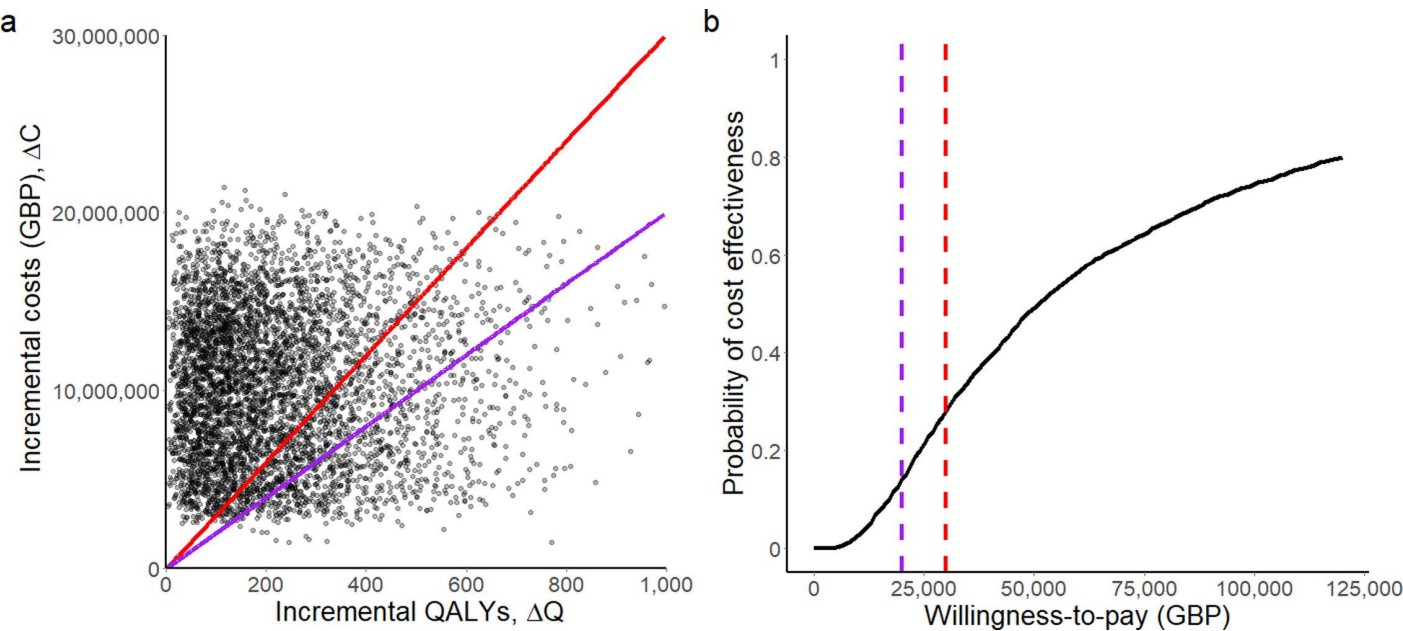

**Fig 1. (a) Incremental cost-effectiveness plane and (b) a cost-effectiveness acceptability curve with a time horizon of 5 years, from the healthcare payer perspective when the cost of shielding is not considered.** (**a**) shows the cost-effectiveness planes of costs based on 5,000 bootstrap replications for 1,000 hypothetical patients. The mean incremental QALYs and incremental costs are approximately 137 and 10M GBP, respectively. The purple and red solid lines are 20,000 and 30,000 GBP per incremental QALY, respectively. (**b**) shows the cost-effectiveness acceptability indicating the probability that the intervention (a mixture of shielding and LAAB-PrEP) is cost effective compared with the standard of care (shielding) for a given willingness-to-pay threshold without shielding costs considered. The y-axis represents the probability that the intervention will be cost effective, and the x-axis represents a range of values for the willingness-to-pay threshold. The purple and red dashed lines represent willingness-to-pay thresholds of 20,000 and 30,000 GBP, respectively. The discounting rate for costs and QALYs is both 3.5%.

more influential for the incremental QALYs and ICERs. Two shielding behaviour-related parameters, i.e., shielding fatigue ($\eta$) and shielding replacement ($\rho$), are very influential for ICERs over a short time-horizon (2 and 5 years), while the life expectancy of immunocompromised individuals in the absence of COVID-19 ($\mu$) and the efficacy of shielding in reducing COVID-19 infections have greater impacts over a longer time horizon, i.e., 10 years.

## Scenario analysis of behavioural parameters

The behavioural parameters are the most uncertain elements in our model. We explored three parameters related to behaviour, as follows. 1) Historical effectiveness of shielding, $\sigma$; this could not be measured due to the lack of a control group, although shielding effectiveness was estimated to be 50% among pregnant women in the USA [25]. 2) Probability of future cessation of shielding regardless of whether a pharmaceutical alternative is available, which we term 'shielding fatigue', $\eta$. This cannot be measured either, as it is a potential future behaviour change. 3) Probability of future cessation of shielding only if provided with LAAB-PrEP, which we term 'replacement', $\rho$. Again, this cannot be measured as it is a potential future behaviour change. We conducted sensitivity analyses for these three parameters by randomly sampling other parameters from the entire range while constraining these three parameters close to the upper and lower bounds, as shown in Fig 3a (see also Fig Ea in S1 Text where shielding cost is considered). Our analysis showed a future scenario

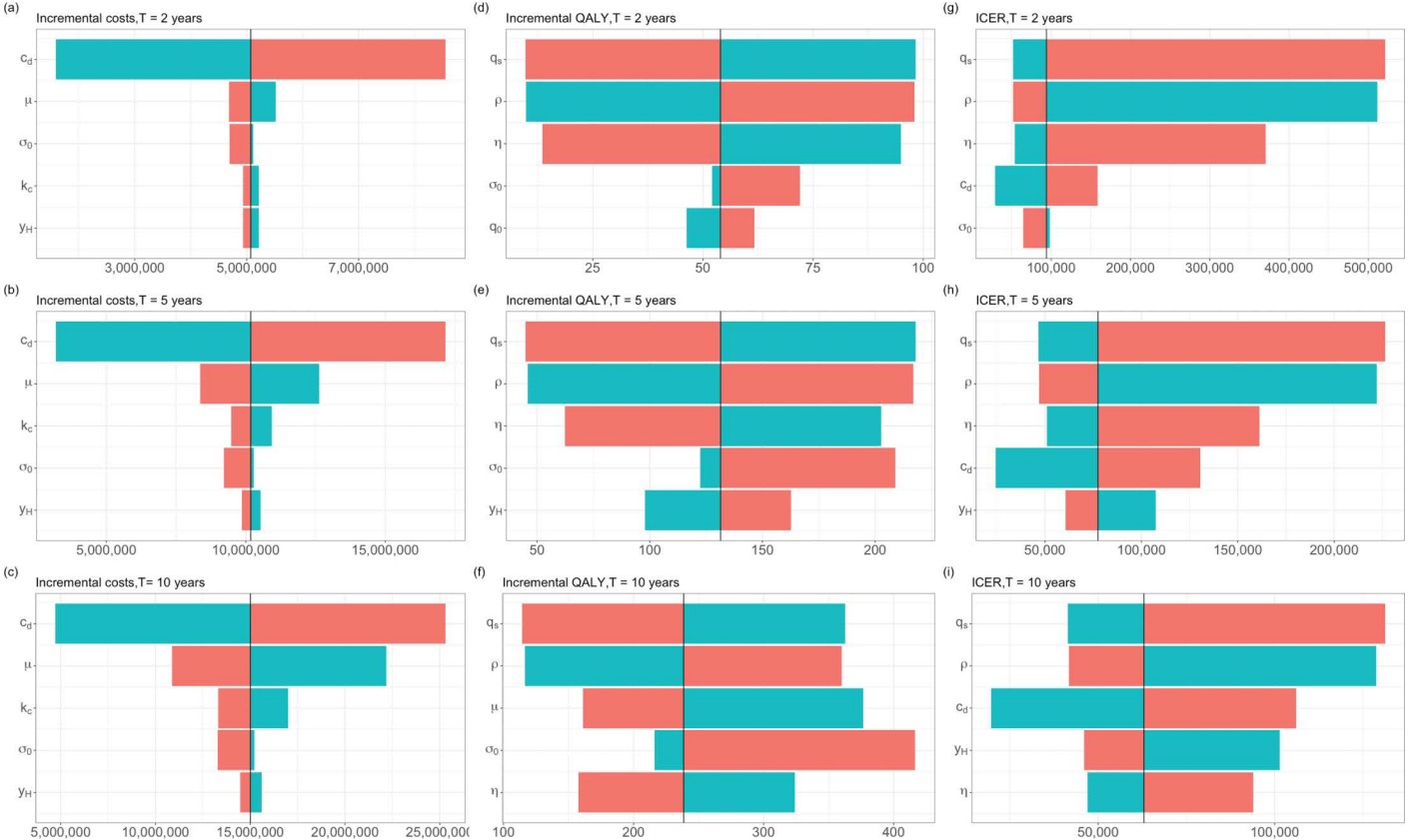

**Fig 2. Tornado plots for the one-way sensitivity analysis between three model outputs (incremental costs, incremental QALYs and ICERs) and the top five features influencing the health and economic impact of the LAAB-PrEP over time-horizons of 2 years (a, d, g), 5 years (b, e, h) and 10 years (c, f, i), from the healthcare payer perspective.** The black vertical lines in each panel represent the outputs for the set of mean values of the parameters. The blue and red bars for each labelled parameter indicate the output where the corresponding parameter was respectively lower and higher than the mean value of its range, while all the remaining parameters were sampled from their entire possible ranges.

where most patients would continue shielding when LAAB-PrEP is not available but would switch to LAAB-PrEP when available (low fatigue, high replacement scenario) represents the future scenario in which LAAB-PrEP is most likely to be cost effective. A future scenario where patients would continue shielding when LAAB-PrEP is not available and would reject LAAB-PrEP when available (low fatigue, low replacement scenario) is the future scenario in which LAAB-PrEP is least likely to be cost effective. The two-way sensitivity analyses, shown in Fig 4, demonstrated how the probability of being cost effective varied conditionally on future LAAB-PrEP products having a particular efficacy and becoming available at a certain price under the three scenarios defined in Fig 3a. This highlighted the strength of the disutility of shielding

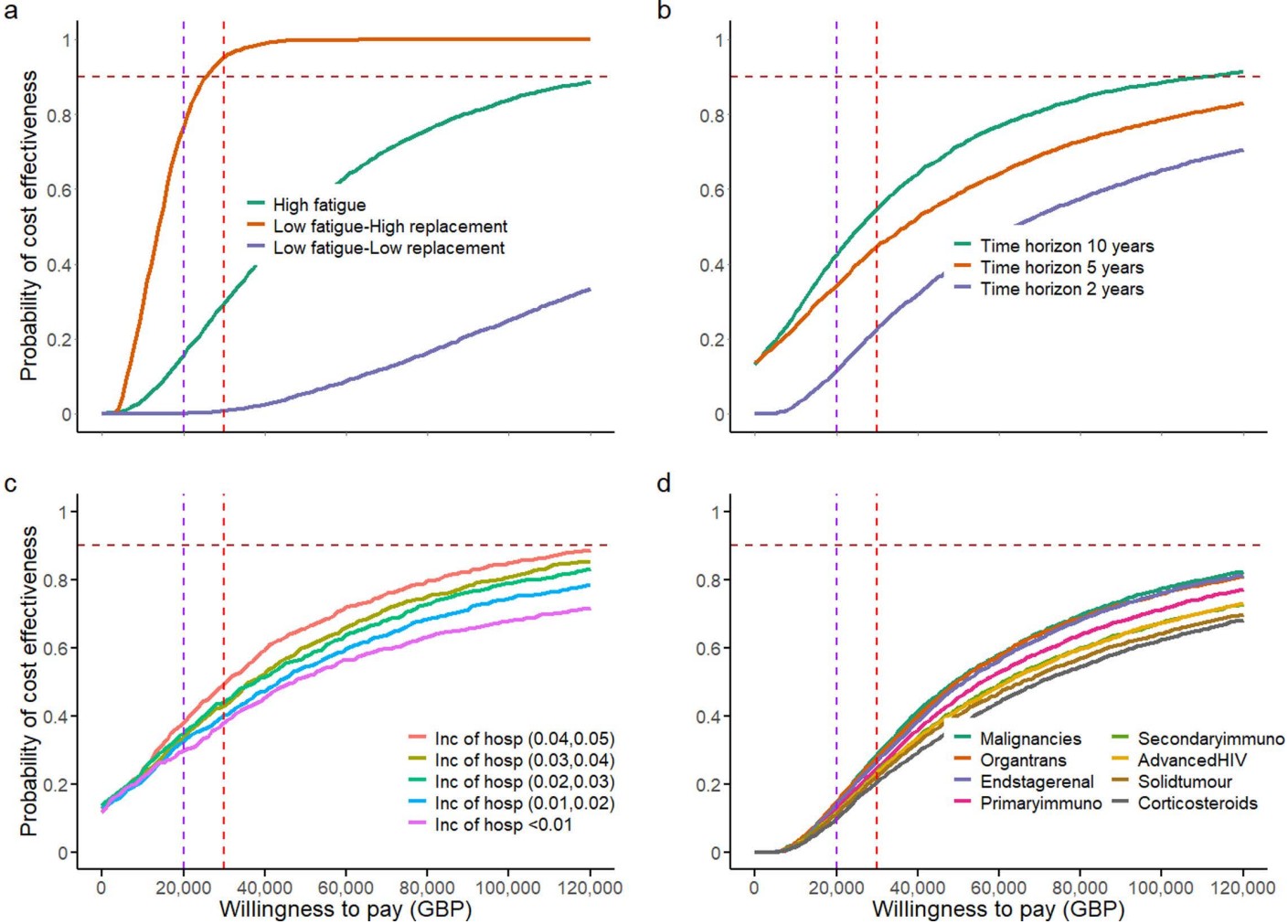

**Fig 3. Cost-effectiveness acceptability curves.** The vertical purple and red dashed lines show the willingness-to-pay thresholds of 20,000 GBP and 30,000 GBP, respectively. The horizontal brown dashed lines show the probability of a cost effectiveness of 0.9. (a) Scenario analysis of behavioural parameters. The low fatigue, high replacement scenario is represented by the red line ($\eta \in [0, \ 0.10]$ and $\rho \in [0.95, \ 1]$). The high fatigue scenario is represented by the green line ($\eta \in [0.75, \ 0.85]$). The low fatigue, low replacement scenario is represented by the purple line ($\eta \in [0, \ 0.15]$ and $\rho \in [0, \ 0.1]$). (b) Sensitivity analysis of time horizons. The green, red and purple lines represent 10, 5 and 2 years, respectively. (c) Sensitivity analysis of the incidence of hospitalisation after COVID-19 infection for individuals who are immunocompromised. The red, yellow, green, blue and pink lines represent the intervals of incidence of hospitalisation as (0.04,0.05), (0.03,0.04), (0.02,0.03), (0.01,0.02) and <0.01 per person per year, respectively. (d) Sensitivity analysis of risk subgroups. The legends in (d) (from left to right) are ordered from the top line to the bottom line. Details of the characteristics and parameters of each group are described in Table C in S1 Text and Evans et al. (2023) [6].

and thus the strength of behaviour changes in influencing the cost effectiveness of LAAB-PrEP. We found that the effectiveness of LAAB-PrEP would only have a small impact on whether its use is cost effective. This is because the majority of the benefit would be due to behaviour change (reduced shielding), with the direct protective effect against COVID-19 contributing less.

Note that we did not explicitly include factors influencing shielding fatigue and replacement behaviours. For example, the perception of a heightened risk of severe of COVID-19, which could be linked to increasing reports of cases or concerns about an individual's personal risk, would reduce fatigue. A reduced confidence in the ability of LAAB-PrEP to provide sufficient protection, which could be linked to reports on its effectiveness against circulating variants, would reduce shielding replacement.

### Lengthening the time horizon increases cost effectiveness

Running the repeated dosing and repeated challenge programme for a longer time horizon, e.g., 10 years, would increase the total costs but gain additional QALYs, resulting in a decreasing cost per QALY gained (Fig 2). The longer the time horizon, the more likely that LAAB-PrEP will be cost effective (Fig 3b; see also Fig Eb in S1 Text where shielding cost is considered). Interestingly, the relative importance of various parameters differs when considering different time-horizons, especially for QALYs and ICERs (Fig 2). When the time horizon is short (2 or 5 years), the impact of shielding on the utility value of an immunocompromised individual's life ($u_s$), together with other parameters relevant to shielding behaviour, plays an important role in determining the QALYs of LAAB-PrEP, whereas life expectancy, $\frac{1}{\mu}$, becomes more important when the time horizon is long (10 years) (Fig 2). Intuitively, we consider this to be because of the competing risk between the mortality rate in the absence of COVID-19 infection among immunocompromised individuals and the

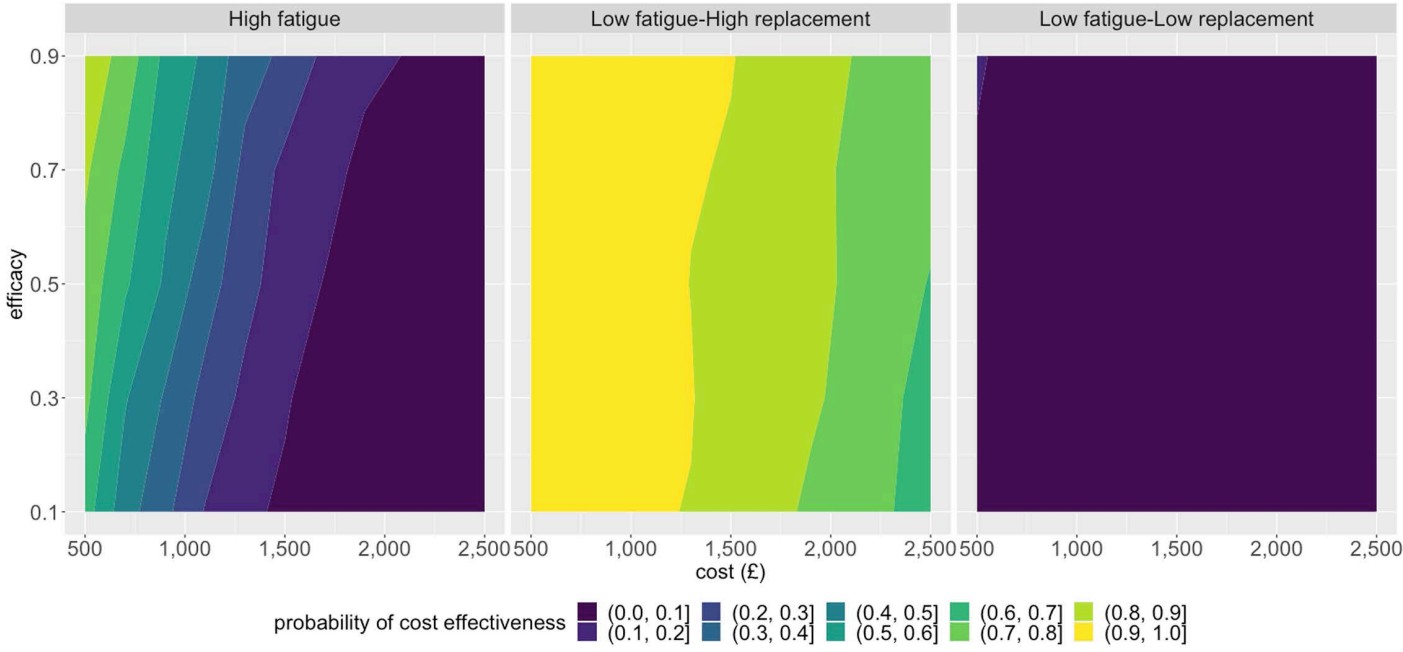

**Fig 4. The two-way sensitivity analysis for the probability of cost effectiveness varied conditionally according to whether future LAAB-PrEP products have a particular efficacy and if they become available at a certain price under the three scenarios studied in Fig 3a.** Specifically, the "high fatigue" scenario is defined by values of $\eta \in [0.75,\ 0.85]$, the "low fatigue high replacement" scenario is defined by values $\rho \in [0.95,\ 1]$ and $\eta \in [0,\ 0.15]$, and the "low fatigue-low replacement" scenario is defined by values $\rho \in [0,\ 0.1]$ and $\eta \in [0, 0.15]$. For this analysis, the willingness-to-pay threshold was fixed at GBP 20,000.

COVID-19 mortality rate. LAAB-PrEP will only be cost effective if the risk of mortality due to COVID-19 infection is higher than the mortality rate of comorbidity in the absence of COVID-19. This competing risk is more pronounced with a longer time-horizon.

**Exploring the probability of LAAB-PrEP cost-effectiveness under different incidences of COVID-19 hospitalisation**

As the incidence of hospitalisation can be measured and varies greatly among individuals who are immunocompromised, we also explored the probability of the cost effectiveness of LAAB-PrEP for different ranges of incidence of COVID-19 hospitalisation. We found that a higher incidence of hospitalisation leads to a higher probability of LAAB-PrEP being cost effective (Fig 3c; see also Fig Ec in S1 Text where shielding cost is considered), implying that targeting individuals with a higher risk of severe COVID-19 might help to optimise the overarching cost-effectiveness of the rollout. This was confirmed by Fig 3d (see also Fig Eb in S1 Text where shielding cost is considered), where we parameterised eight risk subgroups with a variety of primary conditions (as defined in the INFORM study [6] and summarised in our Table C in S1 Text). These different risk groups were parameterised with different incidence of COVID-19 hospitalisation. The risk groups also had different values for the probability of requiring general hospital bed treatment given symptoms, the probability of requiring intensive care unit (ICU) without ventilator treatment given hospitalisation, and the average time to death in the absence of COVID-19.

## Discussion

Here, we present a modelling framework to evaluate which features of a hypothetical pharmaceutical intervention (LAAB-PrEP), in combination with shielding, have the greatest influence on the health and economic impacts of that intervention in England or a similar setting. The target population for this intervention is individuals who are immunocompromised and thus at high risk of severe COVID-19 and its complications. The current standard of care for COVID-19 prophylaxis in this risk group is shielding, a non-pharmaceutical intervention. As noted earlier, creating a single framework to evaluate and compare pharmaceutical and non-pharmaceutical interventions is challenging. If a constant effectiveness is maintained, we predict that LAAB-PrEP taken over a longer time-horizon as a repeated dosing prophylaxis strategy against repeated COVID-19 challenges is more cost effective than a short-term, temporary strategy. The influence of dynamically varying effectiveness is the subject of the current study.

A cost-effectiveness analysis of *Evusheld* PrEP for individuals who are immunocompromised has been conducted in South Korea [26]. In that study, a transmission model was used to estimate the incremental cost-effectiveness (cost per QALY gained) of *Evusheld* from the healthcare system perspective. However, there are some important caveats to note with that study compared with the approach we have taken. Jo and colleagues' model does not consider reinfection or repeated dosing, which might lead to a biased estimate of the impact of LAAB-PrEP. Also, they studied a 1-year time-horizon, whereas we considered a repeated challenge and repeated dosing strategy in our model and conducted a sensitivity analysis for time horizons of 2, 5 and 10 years. Furthermore, Jo and colleagues' comparator was the same population but without *Evusheld*. This overlooks the primary standard of care among immunocompromised individuals, i.e., shielding. In our model, we compare LAAB-PrEP with shielding, specifically using parameters based on the costs of shielding in England (which may be higher or lower for the individual and/or state in other settings). Finally, they considered the risk of COVID-19 infection to be the same among immunocompromised individuals and the general population, which may not be the case and which we attempted to account for in our approach.

Some attempts have been made to estimate the effectiveness of shielding among individuals assumed to be at high risk of severe COVID-19, by comparing COVID-19 outcomes among the shielding, high-risk population and the non-shielding, general population [10,27]. However, this comparison overlooks the fact that the non-shielding, general population has very different susceptibility and immune responses before [28–31] and after vaccination [32] compared with the shielding, high-risk population. This leads to a biased estimate of the effectiveness of shielding at reducing the

incidence of infection and moderating disease severity. Ideally, the analysis should be COVID-19 outcomes among a shielding high-risk population compared with a non-shielding high-risk population. However, this would be unethical and also impractical, given that all individuals in high-risk groups were advised to shield during the pandemic.

Additionally, such a comparison would normally involve a comparator that is a pharmaceutical intervention and an intervention that is also a (potentially superior) pharmaceutical intervention. If the new pharmaceutical intervention was indeed superior, it is assumed that the entire target population would switch from the old (comparator) intervention to the new one. In our study, however, the comparator (shielding) is a non-pharmaceutical intervention. Members of the target population could decide to take the pharmaceutical intervention (in this case LAAB-PrEP) but subsequently all of these individuals, none of them, or some fraction in between could do both, i.e., take LAAB-PrEP while continuing to shield. Future scenarios may not necessarily therefore involve a complete replacement of shielding but could be a variety of combinations of current or reduced levels of shielding combined with uptake of the new pharmaceutical intervention (LAAB-PrEP).

This study has a number of limitations. 'Shielding' behaviour of individuals who are immunocompromised could provide them with protection against a wide range of pathogens beyond SARS-CoV-2, which was not considered in this analysis. This might have resulted in shielding effectiveness from a societal perspective being underestimated. We did not consider the impact of long COVID, which could be prevalent in these immunocompromised individuals; this would lead to a disutility associated with COVID-19 infection, and the exclusion of this in the model would mean that the cost-effectiveness of preventing infection (through pharmaceutical or non-pharmaceutical interventions) is underestimated. We did not include ageing or age-stratified mortality in the model. Mortality is, however, stratified by non-COVID mortality versus COVID-related mortality. Age is less relevant for the immunocompromised risk group because their mortality rate is more closely linked with their underlying primary condition. In the individual risk-group analysis (Fig 3d), we did not include the group-specific utility value of the underlying morbidity because the relevant estimates were not to our knowledge available in the literature, highlighting the need for future research to address this knowledge gap. We found shielding behaviour to be highly influential in terms of the likelihood of LAAB-PrEP being cost effective. However, there is considerable uncertainty around the features of shielding, largely because these features are difficult to measure, such as the cost and effectiveness of shielding in the past. There are also parameters that could not be measured, such as the expected future degree of shielding among individuals who are immunocompromised, because they are related to future behaviours and risk preferences.

When considering the effect of shielding, it is important to note that the incidence of COVID-19 infections and hospitalisations among immunocompromised individuals in the past was measured at the peak of the pandemic, when the advice to shield was in place. This implies that if shielding stops at some point in the future having been effective in the past, then input values for future incidence should be inflated accordingly.

Our modelling framework allowed us to evaluate the QALYs lost due to COVID-19 over various time-horizons, i.e., 2 years, 5 years (default) and 10 years, from the payer (healthcare sector) perspective [33]. This involves costs to healthcare payers, including hospital treatment costs and therapy costs, and also from the societal perspective, involving the approximate costs of shielding (assumed to be in the range of the furlough payments in England during the COVID-19 lockdown [34] and Statutory Sick Pay (SSP) [35]).

Using our repeated challenge and repeated dosing modelling framework, we observed a trade-off between the risk of dying of COVID-19 versus the risk of dying of a primary condition. This is because LAAB-PrEP will only be cost effective if it can reduce the mortality rate due to COVID-19 to less than the mortality rate due to the primary condition. This effect becomes more pronounced over a longer time-horizon, indicating that repeated boosting with LAAB-PrEP for individuals who are immunocompromised, a strategy similar to repeated vaccine boosters for high-risk non-immunocompromised individuals, is a more cost-effective strategy than considering a single dose over a shorter time-horizon.

Additionally, the incidence of hospitalisation differs for different subgroups of immunocompromised individuals. As this parameter can be directly measured in clinical studies, we have parameterised our model so that incidence of

hospitalisation can be used as a model input. The increased probability of cost effectiveness of LAAB-PrEP in high-risk subgroups with a high incidence of hospitalisation may support a prioritisation strategy for subgroups of individuals who are immunocompromised.

The effectiveness of LAAB-PrEP is ranked at a low position in the list of influential parameters, i.e., although influential, other features may be more so. However, this does mean that there will be parameter combinations in which even LAABs with intermediate effectiveness could be cost effective, especially under the high fatigue and the low fatigue high replacement scenarios (Fig 4). This is important given the uncertainty around the future effectiveness of LAABs against new variants of SARS-CoV-2. However, the likelihood of LAAB-PrEP replacing shielding depends on the confidence individuals have in its protective effect, which may in turn be influenced by its perceived and/or reported effectiveness against circulating variants. The dynamic relationship between the effectiveness of LAAB-PrEP, guidelines and physicians' recommendations, and shielding behaviour change by immunocompromised individuals and their close contacts has not been sufficiently characterised for inclusion in a model. The omission of this potential feedback loop between LAAB-PrEP effectiveness and shielding behaviour change is a limitation of the current approach and is to be the topic of future interdisciplinary research.

By taking shielding behaviour into consideration, we have shown that the expected health impact of LAAB-PrEP, as measured by QALYs gained, on COVID-19 in individuals who are immunocompromised, will be driven by various factors. These are the utility of shielding, the features of current and future shielding behaviour, the incidence of COVID-19 hospitalisation in a specific immunocompromised subgroup, and the effectiveness of LAAB-PrEP in preventing clinically significant disease over a short time-horizon and the additional impact of the underlying mortality rate of the primary condition of a specific immunocompromised subgroup over a long time-horizon. The features influencing the expected economic impact of LAAB-PrEP are the cost of LAAB-PrEP, shielding fatigue, the cost of shielding, and shielding replacement. As for the ICER, shielding-related parameters, including shielding fatigue and replacement, utility and cost of shielding, ranked as the most influential parameters.

## Methods

### Model structure

We developed a compartmental susceptible–infected–susceptible (SIS) model structured to capture the infection dynamics of COVID-19 in individuals who are immunocompromised. A simplified version is shown in Fig 5; see Fig A in S1 Text for the complete version. The model explicitly includes disease states and progress. As individuals are infected, they develop either clinically significant or clinically insignificant disease ($I_C$). The model tracks the healthcare requirements of individuals with clinically significant disease; these include a general hospital bed ($I_H$), an intensive care unit (ICU) bed ($I_U$) or an ICU bed and ventilation ($I_V$). These pathways are distilled into a decision-tree algorithm (Fig B in S1 Text) driven by the variables listed in Table A in S1 Text and the parameters listed in Tables B-D in S1 Text. Disease-induced mortality rates are heavily dependent on the severity of the infection outcome.

### Pre-intervention target group

We created a model for a hypothetical cohort with a population size of 1,000, denoted as $Y$, who are assumed to be the target population for the hypothetical LAAB-PrEP, i.e., individuals who are immunocompromised. However, immunocompromised individuals comprise a heterogeneous population with varying degrees of primary conditions that can not only contribute to disutility in the absence of COVID-19 but also lead to great uncertainties around the risk of infection and severity of COVID-19. In general, we handle these uncertainties by sampling parameters characterising immunocompromised individuals across large ranges (Table B in S1 Text) and conducting the cost-effectiveness analysis based upon the model-estimated incremental cost and effect for each set of input parameter values. Furthermore, we parameterised eight

risk subgroups of immunocompromised individuals, based on broad definitions encompassing all conditions/treatments generally considered to result in an immunocompromised state. These definitions were obtained from Evans et al. (2023) [6] and are summarised in our Table C in S1 Text.

### Characterising the effectiveness of shielding among immunocompromised individuals

We consider shielding behaviour among immunocompromised individuals from three standpoints: the past, present and future. The reason for this approach is that immunocompromised individuals in the future may experience shielding fatigue as a result of shielding during the COVID-19 pandemic, which occurred in the past; furthermore, shielding behaviour in the future may be dependent on the availability of alternative pharmaceutical interventions. Additionally, observed incidence of hospitalisation in the present is calculated from immunocompromised individuals who are under historical/present levels of shielding.

If we assume the current estimate of hospitalisation incidence of immunocompromised individuals due to a COVID-19 infection is $y_H$ and the probability of requiring hospital treatment after COVID-19 infection is $p_H$, then the force of infection among immunocompromised individuals who are shielding is given by

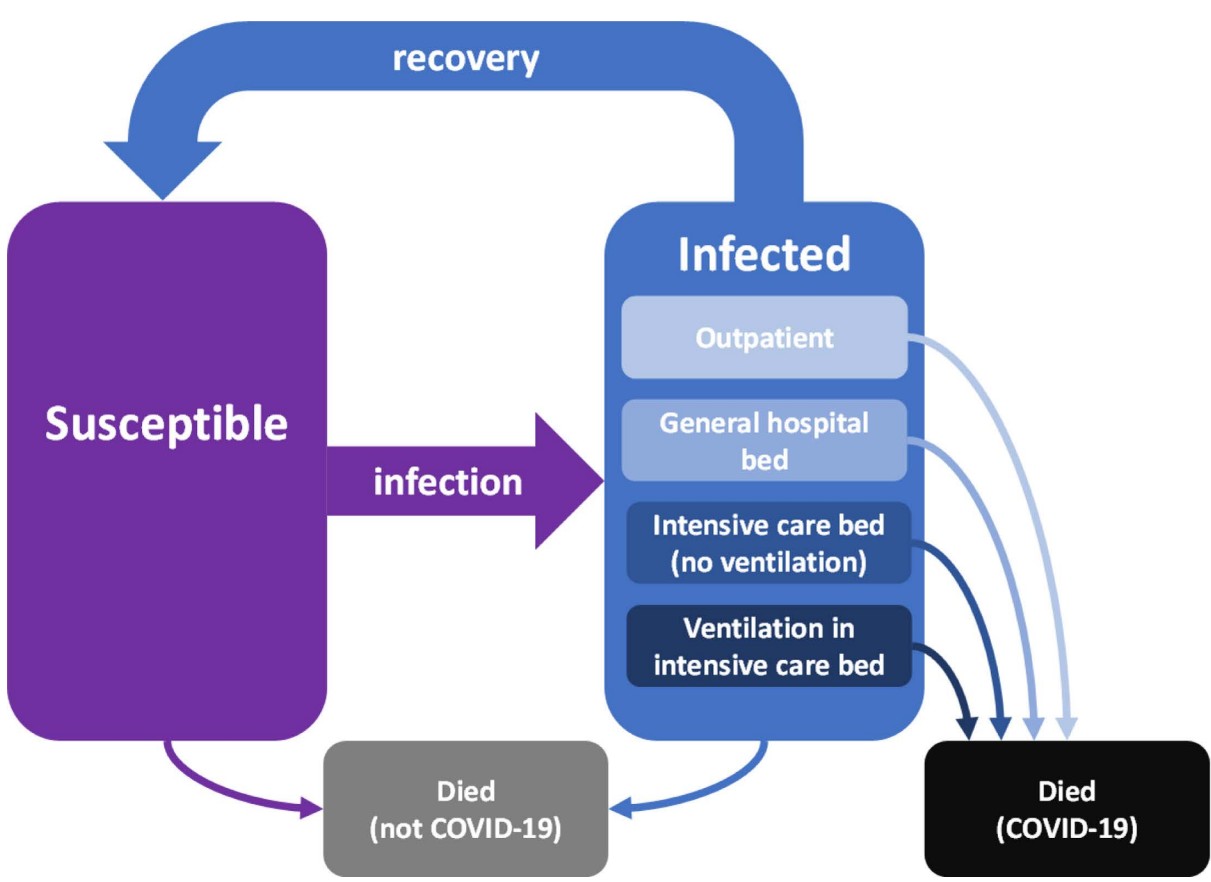

**Fig 5. Flow chart of the susceptible–infected–susceptible (SIS) model showing four severity sub-states for infected individuals: outpatient, hospitalisation requiring a standard hospital bed, hospitalisation requiring an intensive care unit (ICU) bed (no ventilation) and hospitalisation requiring an ICU bed and ventilation.** Death from COVID-19 occurs from the infected states at rates dependent on the level of severity of the infection. Death from other causes, including the primary condition of the immunocompromised individuals, occurs at the same rate for all susceptible and infected states.

$$\lambda_{obs} = \frac{y_H}{p_H} \qquad (1)$$

Then, if they stop the historical/present degree of shielding, the force of infection among individuals who are immuno-compromised, which is an inflated force of infection, can be calculated by

$$\lambda_{inf} = \frac{\lambda_{obs}}{1 - \sigma} \qquad (2)$$

where $\sigma$ denotes the effectiveness of shielding in the past/present.

If we assume the future level of shielding fatigue following the peak of the COVID-19 pandemic that reduces the degree of shielding is $\eta$, the force of infection among immunocompromised individuals in the future in the comparator group is given by

$$\lambda_0 = \lambda_{inf} \times (1 - (1 - \eta) \times \sigma) \qquad (3)$$

If we assume that shielding replacement, defined as the reduction in shielding when an alternative pharmaceutical intervention is available in the future, is $\rho$, then the force of infection among immunocompromised individuals in the future in the intervention group is given by

$$\lambda_1 = \lambda_{inf} \times (1 - (1 - \eta) \times \sigma \times (1 - \rho)) \qquad (4)$$

### Characterising the costs of shielding among immunocompromised individuals

For the weekly shielding cost, $c_s$, we approximated its lower bound based on the Statutory Sick Pay (SSP) level in England, i.e., GBP 99.35 per week for 2022–23 [35], and the upper bound as being analogous to the furlough payments in England; more details are provided in S1 Text.

### Model-estimated incremental costs and incremental effects

The costs of the intervention group (for a mixture of shielding and LAAB-PrEP), $C_1$, comprise three components: hospital treatment costs, $C_{H1}$, shielding (furlough) costs, $C_{s1}$, and the LAAB-PrEP rollout costs, $C_d$, as given by Equation 5 and Equations 1–5 in S1 Text. As we assume the hypothetical LAAB-PrEP is only rolled out every 6 months, the LAAB-PrEP rollout costs are aggregated from surviving individuals (i.e., those who survived COVID-19 infection and any other primary conditions) every 6 months using the price per dose. The costs of the comparator (shielding) group, $C_0$, comprise two components: hospital treatment costs, $C_{H0}$, and shielding (furlough) costs, $C_{s0}$, as given by Equation 6 and Equations 1–4 in S1 Text.

$$C_1 = C_{H1} + C_d + (1 - \rho)(1 - \eta)C_{s1} \qquad (5)$$

$$C_0 = C_{H0} + (1 - \rho)C_{s0} \qquad (6)$$

Here, $C_{H1}$ and $C_{H0}$ are calculated as the product of the length of stay and cost per day for each of the possible types of hospitalisation stay in the intervention group and comparator group, respectively.

We discounted costs of the intervention and comparator groups by applying a discounting parameter, $k_c$, both fixed at 3.5% as recommended by NICE [24] (Table B in S1 Text).

We reviewed relevant literature to estimate a cost per rollout for the hypothetical LAAB-PrEP. *Evusheld* was the first PrEP against COVID-19 (other than vaccination) to receive Emergency Use Authorisation (EUA) and, to the best of our knowledge, to date it remains the only mAb used for PrEP against COVID-19. The list price of one dose (600 mg, containing 300 mg each of tixagevimab and cilgavimab) is GBP 1,600 plus value-added tax (VAT) at 20% [7]. The cost to a private healthcare provider will be GBP 1,200 per dose; however, the cost a recipient pays may be in the order of GBP 2,600 after taking consultation and other fees into account [36]. The cost to public healthcare providers would likely be somewhere between these two costs.

### Calculating differences in costs

Costs were estimated for both the comparator and intervention groups, with the incremental costs, $\Delta C$, therefore being the difference between these estimates.

### Modelling shielding disutility

QALYs were estimated by applying utility decrements to the shielding impact and discounting them from the utility value of primary conditions. If we assume the utility value for immunocompromised individuals with underlying conditions before adjusting for shielding is $u_0$ and the discounting impact of shielding on the utility value is $u_s$, we can define a multipliable relationship between utility value, $U$, and the degree of shielding level, $s$, as defined by Equation 7 [9,37] and shown in Fig C in S1 Text, with the ranges of parameters shown in Table B in S1 Text.

$$U = u_0(1 - (1 - u_s)\,s)$$

(7)

### Differences in disease burden

The QALY is a single indicator that combines the measurement of utility and the length of life [23] and can be calculated using Equations 8 and 9:

$$QALY_0 = (u_0\,(1 - (1 - u_s)\,(1 - \eta)))\,LY_0$$

(8)

$$QALY_1 = (u_0\,(1 - (1 - u_s)\,(1 - \eta)(1 - \rho)))\,LY_1$$

(9)

Here, $QALY_0$ and $QALY_1$ are the QALYs for individuals in the comparator and intervention groups, respectively. $LY_0$ and $LY_1$ are the life-years for individuals in the comparator and intervention groups, respectively; they can be calculated using Equations 8 and 9 in S1 Text.

The incremental QALYs ($\Delta Q$), therefore, are the difference between $QALY_0$ and $QALY_1$.

### Combining costs and effects

Once the incremental costs, $\Delta C$, and incremental QALYs, $\Delta Q$, have been computed, then the ratio, $\Delta C / \Delta Q$, is the incremental cost-effectiveness ratio, ICER. This can be used to determine whether the alternative intervention (mixture of LAAB-PrEP and shielding) is more cost effective than the current standard of care, i.e., shielding.

### Constructing the comparator and the intervention

First, we create a hypothetical population by sampling parameters characterising immunocompromised individuals. Then, we simulate the five infection stages, including the feature of reinfection, among these populations using our SIS

model (Fig 5) and by sampling epidemiological parameters. The hospital treatment costs and LAAB-PrEP rollout costs can be simultaneously aggregated for all infection stages by the time horizon (5 years as the default) that we are interested in.

In the comparator group where the standard of care is shielding with some level of fatigue, historical/current shielding effectiveness and future fatigue level are used to inflate the costs and effects of shielding. In the intervention group, which is a combination of LAAB-PrEP and shielding, we use the replacement level of shielding to further adjust the effect of the comparator group to account for potential changes in the population's preferences should an alternative pharmaceutical intervention become available in the future. The sampling algorithm is illustrated in Fig 6.

### Exploring uncertainty

As already noted, our model can take parameter uncertainties into account. We performed sensitivity and scenario analyses to explore the implications of these parameter uncertainties. For the sensitivity analyses, the uncertainty is captured by randomly drawing values from the cost and effect distributions of all interventions and basing the calculations on these randomly drawn values. First, we conducted a probabilistic sensitivity analysis and output 100,000 samples. Each sample takes in a specific combination of parameters and outputs the corresponding incremental QALYs ($\Delta Q$), incremental costs ($\Delta C$) and ICER. We rank the parameters by their Pearson correlation coefficients with $\Delta Q$ to evaluate the importance of parameters influencing incremental QALYs.

For the scenario analyses, we defined three scenarios by changing the ranges of two parameters relating to the behaviour of immunocompromised patients, including future cessation of shielding regardless of whether a pharmaceutical alternative is available, termed 'shielding fatigue', and future cessation of shielding only if provided with LAAB-PrEP, termed 'shielding replacement'. These scenarios were selected in a purposeful way to maximise the likelihood of a cost-effective outcome. In the low fatigue, low replacement scenario, both values are uniformly sampled between 0 and 0.25. In the low fatigue, high replacement scenario, shielding fatigue is uniformly sampled between 0 and 0.25 and

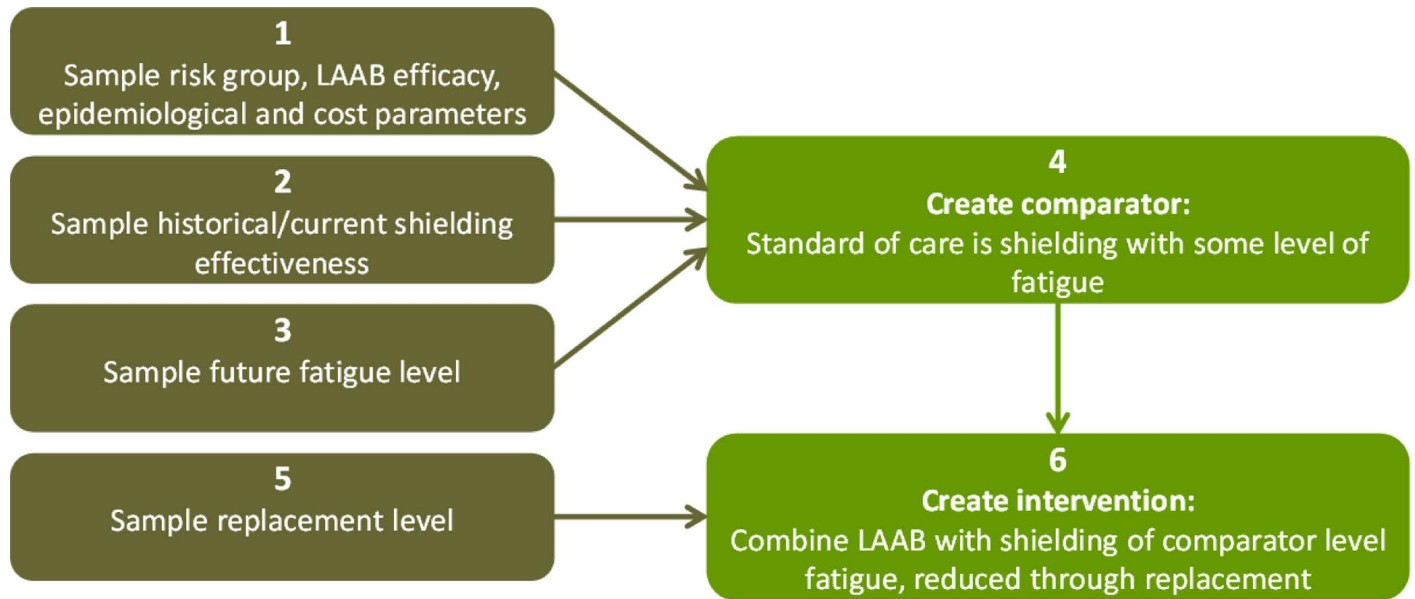

**Fig 6. A conceptual diagram for constructing the comparator (shielding) and the intervention (a combination of shielding and LAAB-PrEP) in the simulation model.**

shielding replacement is uniformly sampled between 0.75 and 0.9. In the high fatigue scenario, shielding fatigue is uniformly sampled between 0.75 and 1 and shielding replacement is uniformly sampled between 0 and 0.9.

In the sensitivity analysis for time horizons, all parameters are randomly sampled from their distributions listed in Table B in S1 Text except the time horizon, which is fixed at 2, 5 or 10 years (Fig 2b). In the sensitivity analysis for the incidence of hospitalisation with COVID-19 infection, after all parameters are randomly sampled from their distributions listed in Table B in S1 Text, the pairs of probabilities of cost effectiveness and incidence of hospitalisation are rearranged so that different lines of probability of cost effectiveness against different intervals of incidence of hospitalisation can be visualised, as shown in Fig 2c. In the sensitivity analysis for individual risk subgroups, parameters characterising the eight risk subgroups were obtained from the INFORM study [6] and Table B in S1 Text, while other parameters were randomly sampled from the distributions listed in Table B in S1 Text as shown in Fig 2d.

## Supporting information

**S1 Text. Supplementary Information.** Fig A. Diagram of the model and the relationships between variables and parameters. Fig B. Diagram showing a probability tree for COVID-19 disease progression. Fig C. A mathematical relationship between utility and the degree of shielding. Fig D. (a) Incremental cost-effectiveness plane and (b) a cost-effectiveness acceptability curve with a time horizon of 5 years when the shielding cost is considered. Fig E. Cost-effectiveness acceptability curves when the cost of shielding is included. Fig F. One-way sensitivity analysis between incremental costs, incremental QALYs and ICER and features influencing the health and economic impact of the LAAB-PrEP over different time-horizons of 2 years (a,d, g), 5 years (b, e, h) and 10 years (c, f, i) when the shielding cost is considered. Fig G. Correlations between incremental QALYs and features influencing the health impact of the LAAB-PrEP and shielding combination based on different time-horizons of 2 years (a), 5 years (b) and 10 years (c). Table A. Definitions of the variables used in the model structure in Figs A and B in S1 Text. Table B. Definitions of the parameters used for the calculations and, where relevant, the distributions used for the probabilistic sensitivity analysis. Table C. Summary of parameters in risk subgroups of immunocompromised individuals, as defined by the INFORM study. Table D. Definitions of parameters used to calculate the upper bound of the weekly cost of shielding.
(DOCX)

**S1 PRISMA Checklist.  CHEERS reporting checklist for economic evaluations.**
(DOCX)

## Author contributions

**Conceptualization:** Siyu Chen, Sudhir Venkatesan, Klas Bergenheim, Sabada Dube, Wilhelmine Meeraus, Cátia Ferreira, Sylvia Taylor, Lisa J White.

**Data curation:** Siyu Chen, Lisa J White.

**Formal analysis:** Siyu Chen, Lisa J White.

**Funding acquisition:** Lisa J White.

**Investigation:** Siyu Chen, Sudhir Venkatesan, Sofie Arnetorp, Klas Bergenheim, Sabada Dube, Wilhelmine Meeraus, Cátia Ferreira, Sylvia Taylor, Lisa J White.

**Methodology:** Siyu Chen, Lisa J White.

**Project administration:** Sudhir Venkatesan, Lisa J White.

**Resources:** Siyu Chen, Sudhir Venkatesan, Lisa J White.

**Software:** Siyu Chen, Lisa J White.

**Supervision:** Lisa J White.

**Validation:** Siyu Chen, Lisa J White.

**Visualization:** Siyu Chen, Lisa J White.

**Writing – original draft:** Siyu Chen, Lisa J White.

**Writing – review & editing:** Siyu Chen, Sudhir Venkatesan, Sofie Arnetorp, Klas Bergenheim, Sabada Dube, Wilhelmine Meeraus, Cátia Ferreira, Sylvia Taylor, Lisa J White.

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
