## [Decision Letter · Decision Letter 0]

15 Jul 2024

Dear Miss Chen,

Thank you very much for submitting your manuscript "Features influencing the health and economic impact of long-acting antibody prophylaxis against COVID-19" for consideration at PLOS Computational Biology.

As with all papers reviewed by the journal, your manuscript was reviewed by members of the editorial board and by several independent reviewers. In light of the reviews (below this email), we would like to invite the resubmission of a significantly-revised version that takes into account the reviewers' comments.

We cannot make any decision about publication until we have seen the revised manuscript and your response to the reviewers' comments. Your revised manuscript is also likely to be sent to reviewers for further evaluation.

Sincerely,

Joseph T. Wu

Academic Editor

PLOS Computational Biology

Virginia Pitzer

Section Editor

PLOS Computational Biology

Reviewer's Responses to Questions

**Comments to the Authors:**

Reviewer #1: This is a well written paper describing an interesting analysis.

AUTHOR SUMMARY

Please clarify whether shielding involves wearing a face mask and only going out when essential or if it is something else. I didn’t understand what shielding meant from this section.

INTRODUCTION

This is well written and does a good job of setting the scene for the analysis.

RESULTS

Fig 1a – suggest making the dots smaller and/or a lighter color so you can tell where multiple iterations stack on top of each other. y-axis appears to be wrong (incremental costs of ~$15m)

Lines 244-247 - I don’t think I follow what this sentence is saying

Fig 2 – Initially found this strange because the text on 2a and 2c isn’t mentioned until the paragraph beginning at line 280. Is it possible to shift the order around so that this is covered earlier?

fig 3d is hard to read. Suggest changing colors or making larger.

DISCUSSION

Does the paragraph in lines 373-378 relate at all to the study from South Korea mentioned in lines 316-331?

METHODS

Clearly state the perspective for the analysis. Lines 381=382 state that you’re taking the patient and the healthcare sector perspectives—do you mean two different perspectives or the societal perspective?

Since this is set in the UK, suggest using a discount rate of 3.5% in the base case to align with NICE recommendations.

Line 519 – Are you assuming that this drug would only be available in the private sector? What if it became available in the public sector?

Line 526-531 – could you provide further justification for these choices? Is it fair to assume that this is a linear relationship?

Do you include disutility for COVID or just for shielding? Where are the utilities for COVID illness?

SUPP

Table S3 – It would be helpful for the column headers to be explained

Reviewer #2: Review was uploaded as an attachment.

**Have the authors made all data and (if applicable) computational code underlying the findings in their manuscript fully available?**

Reviewer #1: None

Reviewer #2: Yes

PLOS authors have the option to publish the peer review history of their article (what does this mean? ). If published, this will include your full peer review and any attached files.

**Do you want your identity to be public for this peer review?** For information about this choice, including consent withdrawal, please see our Privacy Policy .

Reviewer #1: **Yes: ** Angela Devine

Reviewer #2: No
---

## [Decision Letter · Decision Letter 1]

21 Nov 2024

PCOMPBIOL-D-24-00735R1Features influencing the health and economic impact of long-acting antibody prophylaxis against COVID-19PLOS Computational Biology  Dear Dr. Chen, Thank you for submitting your manuscript to PLOS Computational Biology. After careful consideration, we feel that it has merit but does not fully meet PLOS Computational Biology's publication criteria as it currently stands. Therefore, we invite you to submit a revised version of the manuscript that addresses the points raised during the review process. Please submit your revised manuscript within 30 days Jan 21 2025 11:59PM. If you will need more time than this to complete your revisions, please reply to this message or contact the journal office at ploscompbiol@plos.org.  Please include the following items when submitting your revised manuscript: * A rebuttal letter that responds to each point raised by the editor and reviewer(s). You should upload this letter as a separate file labeled 'Response to Reviewers'. This file does not need to include responses to formatting updates and technical items listed in the 'Journal Requirements' section below. * A marked-up copy of your manuscript that highlights changes made to the original version. You should upload this as a separate file labeled 'Revised Manuscript with Track Changes'. * An unmarked version of your revised paper without tracked changes. You should upload this as a separate file labeled 'Manuscript'. If you would like to make changes to your financial disclosure, competing interests statement, or data availability statement, please make these updates within the submission form at the time of resubmission. Guidelines for resubmitting your figure files are available below the reviewer comments at the end of this letter.We look forward to receiving your revised manuscript.Kind regards, Joseph T. WuAcademic EditorPLOS Computational Biology

Virginia Pitzer

Section Editor

PLOS Computational Biology

Feilim Mac Gabhann

Editor-in-Chief

PLOS Computational Biology

Jason Papin

Editor-in-Chief

PLOS Computational Biology

**Journal Requirements:**

1) Please upload all main figures as separate Figure files in .tif or .eps format. For more information about how to convert and format your figure files please see our guidelines:

2) Please upload a copy of Figure Figure 5 which you refer to in your text on page 27. Or, if the figure is no longer to be included as part of the submission please remove all reference to it within the text.

3) Please amend your detailed Financial Disclosure statement. This is published with the article. It must therefore be completed in full sentences and contain the exact wording you wish to be published.

2) State what role the funders took in the study. If the funders had no role in your study, please state: "The funders had no role in study design, data collection and analysis, decision to publish, or preparation of the manuscript.".

**Reviewers' comments:**

Reviewer's Responses to Questions

**Comments to the Authors:**

Reviewer #1: The authors have addressed my comments

Reviewer #2: Thank you for the opportunity to review this updated manuscript. I have some further questions and comments:

1. The authors mention the NICE highly specialised technologies programme as a potential justification for using a £100,000 per QALY cost-effectiveness threshold. However, this programme is limited to very rare conditions with prevalence below 1 in 50,000 people and an eligible population of 300 – 500 people. From the INFORM study that has been used to guide the authors modelling assumptions, there are roughly 2 million people in England who met the immunocompromised definition. I continue to believe that this threshold does not provide a helpful indication of the potential cost-effectiveness of the proposed application of LAAB-PrEP in the England.

2. In response to my comment that the results presented did not appear to support the likelihood of LAAB-PrEP being cost-effective in the UK, the authors updated their definition of the “low-fatigue high-replacement” scenario so that his now appears to be cost-effective at £20,000 to £30,000 per QALY. I am somewhat uncomfortable with the scenario definition being amended to achieve a particular result. I think as a minimum it should be made clear in the methods that (some of) the scenarios have been selected in a purposeful way (i.e. to maximise the likelihood of a cost-effective outcome.)

3. Given the above, I wonder if the authors might consider presenting the main results in a more theoretical way e.g. by focussing on the multi-way scenario analysis of the key unknown parameters, to highlight how the interaction of these parameters influences potential cost-effectiveness (similar to the new S5 figure.)

4. Related to the above, for the PSA, I am not sure how useful it is to incorporate extremely wide uncertainty e.g. uniform [1/50, 1/5] for death risk, or uniform [0.1,0.9] for LAAB-PrEP into a single simulation. If many parameters are varied over a very wide range that is not informed by empirical data, then the proportion of samples that are cost-effective tell us little about the probability that the intervention will be cost-effective.

5. I was interested to see in figure S5, that the effectiveness of LAAB-PrEP has only a small impact on whether it’s use is cost-effective. This is presumably because the majority of the benefit is down to behaviour change (reduced disutility of shielding) and the direct protective effect against Covid-19 does not actually matter much. In extremis it would seem that much of the benefit could theoretically be achieve by placebo at no cost. In practice presumably individuals would not stop shielding without having good evidence of at least some effectiveness from LAAB-PrEP. Is there any empirical evidence on which the link between risk reduction and shielding behaviour that could be used to inform this link?

**Have the authors made all data and (if applicable) computational code underlying the findings in their manuscript fully available?**

Reviewer #1: None

Reviewer #2: Yes

PLOS authors have the option to publish the peer review history of their article (what does this mean? ). If published, this will include your full peer review and any attached files.

**Do you want your identity to be public for this peer review?** For information about this choice, including consent withdrawal, please see our Privacy Policy .

Reviewer #1: No

Reviewer #2: No

**Figure resubmission:**  While revising your submission, please upload your figure files to the Preflight Analysis and Conversion Engine (PACE) digital diagnostic tool, https://pacev2.apexcovantage.com/. PACE helps ensure that figures meet PLOS requirements. To use PACE, you must first register as a user. Registration is free. Then, login and navigate to the UPLOAD tab, where you will find detailed instructions on how to use the tool. If you encounter any issues or have any questions when using PACE, please email PLOS at figures@plos.org. Please note that Supporting Information files do not need this step. If there are other versions of figure files still present in your submission file inventory at resubmission, please replace them with the PACE-processed versions.**Reproducibility:**  To enhance the reproducibility of your results, we recommend that authors of applicable studies deposit laboratory protocols in protocols.io, where a protocol can be assigned its own identifier (DOI) such that it can be cited independently in the future. Additionally, PLOS ONE offers an option to publish peer-reviewed clinical study protocols. Read more information on sharing protocols at https://plos.org/protocols?utm_medium=editorial-email&utm_source=authorletters&utm_campaign=protocols

---

## [Decision Letter · Decision Letter 2]

10 Apr 2025

PCOMPBIOL-D-24-00735R2

Features influencing the health and economic impact of preventing COVID-19 in immunocompromised individuals

PLOS Computational Biology

Dear Dr. Chen,

Thank you for submitting your manuscript to PLOS Computational Biology. After careful consideration, we feel that it has merit but does not fully meet PLOS Computational Biology's publication criteria as it currently stands. Therefore, we invite you to submit a revised version of the manuscript that addresses the points raised during the review process.

Please submit your revised manuscript within 30 days Jun 10 2025 11:59PM. If you will need more time than this to complete your revisions, please reply to this message or contact the journal office at ploscompbiol@plos.org. Please include the following items when submitting your revised manuscript:

We look forward to receiving your revised manuscript.

Kind regards,

Joseph T. Wu

Academic Editor

PLOS Computational Biology

Virginia Pitzer

Editor-in-Chief

PLOS Computational Biology

**Additional Editor Comments (if provided):**

**Journal Requirements:**

**Reviewers' comments:**

Reviewer's Responses to Questions

**Comments to the Authors:**

Reviewer #2: The authors have addressed the majority of my comments. There is one of my comments from the last review that the authors have not addressed or commented on.

1. In the previous review I wrote that

"The authors mention the NICE highly specialised technologies programme as a potential justification for using a £100,000 per QALY cost-effectiveness threshold. However, this programme is limited to very rare conditions with prevalence below 1 in 50,000 people and an eligible population of 300 – 500 people. From the INFORM study that has been used to guide the authors modelling assumptions, there are roughly 2 million people in England who met the immunocompromised definition. I continue to believe that this threshold does not provide a helpful indication of the potential cost-effectiveness of the proposed application of LAAB-PrEP in the England."

The authors have not addressed this in their response. In my opinion the best option would be to remove this threshold from the figures because for the general reader I think it creates a misleading impression of the likelihood of cost-effectiveness. However, if the authors wish the retain the use of this threshold then I suggest that my point above about the very small population that could potentially be eligible under this scheme is made explicit in both the main text and the figure captions where the threshold is presented.

**Have the authors made all data and (if applicable) computational code underlying the findings in their manuscript fully available?**

Reviewer #2: Yes

PLOS authors have the option to publish the peer review history of their article (what does this mean? ). If published, this will include your full peer review and any attached files.

**Do you want your identity to be public for this peer review?** For information about this choice, including consent withdrawal, please see our Privacy Policy .

Reviewer #2: No

**Figure resubmission:**
---

## [Editor Report · Decision Letter 3]

21 Apr 2025

Dear Dr. Chen,

We are pleased to inform you that your manuscript 'Features influencing the health and economic impact of preventing COVID-19 in immunocompromised individuals' has been provisionally accepted for publication in PLOS Computational Biology.

Best regards,

Joseph T. Wu

Academic Editor

PLOS Computational Biology

Virginia Pitzer

Editor-in-Chief

PLOS Computational Biology

---

## [Editor Report · Acceptance letter]

PCOMPBIOL-D-24-00735R3

Features influencing the health and economic impact of preventing COVID-19 in immunocompromised individuals

Dear Dr Chen,

I am pleased to inform you that your manuscript has been formally accepted for publication in PLOS Computational Biology. Your manuscript is now with our production department and you will be notified of the publication date in due course.

With kind regards,

Anita Estes
